# The Systemic Inflammatory Response Identifies Patients with Adverse Clinical Outcome from Immunotherapy in Hepatocellular Carcinoma

**DOI:** 10.3390/cancers14010186

**Published:** 2021-12-31

**Authors:** Ambreen Muhammed, Claudia Angela Maria Fulgenzi, Sirish Dharmapuri, Matthias Pinter, Lorenz Balcar, Bernhard Scheiner, Thomas U. Marron, Tomi Jun, Anwaar Saeed, Hannah Hildebrand, Mahvish Muzaffar, Musharraf Navaid, Abdul Rafeh Naqash, Anuhya Gampa, Umut Ozbek, Junk-Yi Lin, Ylenia Perone, Bruno Vincenzi, Marianna Silletta, Anjana Pillai, Yinghong Wang, Uqba Khan, Yi-Hsiang Huang, Dominik Bettinger, Yehia I. Abugabal, Ahmed Kaseb, Tiziana Pressiani, Nicola Personeni, Lorenza Rimassa, Naoshi Nishida, Luca Di Tommaso, Masatoshi Kudo, Arndt Vogel, Francesco A. Mauri, Alessio Cortellini, Rohini Sharma, Antonio D’Alessio, Celina Ang, David J. Pinato

**Affiliations:** 1Department of Surgery & Cancer, Imperial College London, Hammersmith Hospital, Du Cane Road, London W12 0HS, UK; ambreen.muhammed17@imperial.ac.uk (A.M.); c.fulgenzi@unicampus.it (C.A.M.F.); y.perone15@imperial.ac.uk (Y.P.); f.mauri@imperial.ac.uk (F.A.M.); a.cortellini@imperial.ac.uk (A.C.); r.sharma@imperial.ac.uk (R.S.); a.dalessio@imperial.ac.uk (A.D.); 2Department of Medical Oncology, University Campus Bio-Medico, 00128 Rome, Italy; b.vincenzi@unicampus.it (B.V.); m.silletta@unicampus.it (M.S.); 3Department of Medicine, Division of Hematology/Oncology, Tisch Cancer Institute, Mount Sinai Hospital, New York, NY 10029, USA; sirish.dharmapuri@mountsinai.org (S.D.); thomas.marron@mssm.edu (T.U.M.); Tomi.Jun@mountsinai.org (T.J.); celina.ang@mssm.edu (C.A.); 4Department of Internal Medicine III, Division of Gastroenterology & Hepatology, Medical University of Vienna, 1090 Vienna, Austria; matthias.pinter@meduniwien.ac.at (M.P.); lorenz.balcar@meduniwien.ac.at (L.B.); bernhard.scheiner@meduniwien.ac.at (B.S.); 5Division of Medical Oncology, Department of Medicine, Kansas University Cancer Center, Westwood, KS 66205, USA; asaeed@kumc.edu (A.S.); hhildebrand@kumc.edu (H.H.); 6Division of Hematology/Oncology, East Carolina University, 600 Moye Boulevard, Greenville, NC 27834, USA; muzaffarm@ecu.edu (M.M.); navaidm@ecu.edu (M.N.); abdulrafeh-naqash@ouhsc.edu (A.R.N.); 7Section of Gastroenterology, Hepatology & Nutrition, The University of Chicago Medicine, 5841 S. Maryland Ave, Chicago, IL 60637, USA; anuhya.gampa@uchospitals.edu (A.G.); apillai1@medicine.bsd.uchicago.edu (A.P.); 8Department of Population Health Science and Policy, Tisch Cancer Institute, Icahn School of Medicine at Mount Sinai, New York, NY 10029, USA; umut.ozbe@mountsinai.org (U.O.); joyceleaves@gmail.com (J.-Y.L.); 9Department of Gastroenterology, Hepatology & Nutrition, The University of Texas MD Anderson Cancer Center, Houston, TX 77030, USA; ywang59@mdanderson.org; 10Division of Hematology and Oncology, Weill Cornell Medicine/New York Presbyterian Hospital, 1305 York Avenue, Room Y1247, New York, NY 10021, USA; uqk9001@med.cornell.edu; 11Division of Gastroenterology and Hepatology, Taipei Veterans General Hospital, Institute of Clinical Medicine, National Yang Ming Chiao Tung University School of Medicine, Taipei 11217, Taiwan; yhhuang@vghtpe.gov.tw; 12Department of Medicine II, Faculty of Medicine, Medical Center University of Freiburg, University of Freiburg, 79106 Freiburg, Germany; dominik.bettinger@uniklinik-freiburg.de; 13Department of Gastrointestinal Medical Oncology, The University of Texas MD Anderson Cancer Center, Houston, TX 77030, USA; yimohamed@mdanderson.org (Y.I.A.); akaseb@mdanderson.org (A.K.); 14Medical Oncology and Hematology Unit, Humanitas Cancer Center, IRCCS Humanitas Research Hospital, Via Manzoni 56, Rozzano, 20089 Milan, Italy; tiziana.pressiani@cancercenter.humanitas.it (T.P.); nicola.personeni@hunimed.eu (N.P.); lorenza.rimassa@hunimed.eu (L.R.); 15Department of Biomedical Sciences, Humanitas University, Via Rita Levi Montalcini 4, Pieve Emanuele, 20072 Milan, Italy; 16Department of Gastroenterology and Hepatology, Kindai University Faculty of Medicine, Osaka-Sayama, Osaka 589-8511, Japan; naoshi@kuhp.kyoto-u.ac.jp (N.N.); m-kudo@med.kindai.ac.jp (M.K.); 17Humanitas Clinical and Research Center IRCCS, Pathology Unit, 20089 Rozzano, Italy; luca.di_tommaso@hunimed.eu; 18Department of Gastroenterology, Hepatology and Endocrinology, Hannover Medical School, 30625 Hannover, Germany; vogel.arndt@mh-hannover.de

**Keywords:** hepatocellular carcinoma, inflammatory biomarkers, neutrophil-lymphocyte ratio, platelet-lymphocyte ratio, prognostic nutritional index

## Abstract

**Simple Summary:**

The investigation of predictive and prognostic markers is pivotal in patients affected by hepatocellular carcinoma treated with immune-checkpoint-inhibitors. Inflammation has a central role in hepatocellular carcinoma development and progression; however, its role in influencing outcomes in the context of immunotherapy has not been fully elucidated yet. In the following study, we investigated the prognostic role of bloods derived inflammatory markers and we found that they predict survival and response of patients treated with immunotherapy for advanced hepatocellular carcinoma.

**Abstract:**

Systemic inflammation is a hallmark of cancer, and it has a pivotal role in hepatocellular carcinoma (HCC) development and progression. We conducted a retrospective study including 362 patients receiving immune check-point inhibitors (ICIs) across three continents, evaluating the influence of neutrophiles to lymphocytes ratio (NLR), platelets to lymphocytes ratio (PLR), and prognostic nutritional index (PNI) on overall (OS), progression free survival (PFS), and radiologic responses. In our 362 patients treated with immunotherapy, median OS and PFS were 9 and 3.5 months, respectively. Amongst tested inflammatory biomarkers, patients with NLR ≥ 5 had shorter OS (7.7 vs. 17.6 months, *p* < 0.0001), PFS (2.1 vs. 3.8 months, *p* = 0.025), and lower objective response rate (ORR) (12% vs. 22%, *p* = 0.034); similarly, patients with PLR ≥ 300 reported shorter OS (6.4 vs. 16.5 months, *p* < 0.0001) and PFS (1.8 vs. 3.7 months, *p* = 0.0006). NLR emerged as independent prognostic factors for OS in univariate and multivariate analysis (HR 1.95, 95%CI 1.45–2.64, *p* < 0.001; HR 1.73, 95%CI 1.23–2.42, *p* = 0.002) and PLR remained an independent prognostic factor for both OS and PFS in multivariate analysis (HR 1.60, 95%CI 1.6–2.40, *p* = 0.020; HR 1.99, 95%CI 1.11–3.49, *p* = 0.021). Systemic inflammation measured by NLR and PLR is an independent negative prognostic factor in HCC patients undergoing ICI therapy. Further studies are required to understand the biological mechanisms underlying this association and to investigate the predictive significance of circulating inflammatory biomarkers in HCC patients treated with ICIs.

## 1. Introduction

Hepatocellular carcinoma (HCC) is the fourth most common cause of cancer-related deaths worldwide [1] with only 24% of patients being disease-free at 5 years, even after curative resection [2]. Immunotherapy with immune checkpoint inhibitors (ICIs) has led to transformative changes in the management of multiple oncological indications [3]. However, wide heterogeneity exists across and within oncological indications and identification of predictive correlates of response and survival benefit is an area of high unmet need in the clinical delivery of immunotherapy. ICIs target inhibitory pathways such as the programmed cell death protein-1/programmed cell death ligand-1 (PD-1/PD-L1) and CTLA-4/CD80/CD86 axes. The forerunners PD-1 blocking antibodies nivolumab and pembrolizumab have demonstrated a response in a little less than 20% of patients with HCC [4], and have yielded controversial results in phase III clinical trials. In fact, nivolumab failed to significantly improve survival compared to sorafenib in the first-line CheckMate 459 study [5] and pembrolizumab was not demonstrated to perform better than placebo in the second-line Keynote-240 clinical trial [6], but it has recently been reported to significantly improve OS, PFS, and ORR compared to placebo in Asian patients treated after sorafenib failure within the phase III Keynote-394 trial (NCT03062358). More recently the combination of the PD-L1 antibody atezolizumab and the VEGF inhibitor bevacizumab demonstrated significant improvement in overall survival (OS) compared to sorafenib, changing the treatment landscape of patients with advanced HCC [7], leading to an unprecedented median OS of 19.2 months [8,9]. Contrary to other malignancies, the development of ICIs for HCC therapy has been predominantly empirical as no biomarkers are available to predict which patients are resistant to ICI monotherapy or likely to respond [3,10]. Moreover, prognostic biomarkers which are established in other oncological indications, such as PD-L1 expression [11] and tumour mutational burden, are not predictive of outcome in HCC [12]. Given the crucial role of inflammation in HCC pathogenesis [13,14], and in the mechanism of action of ICIs, phenotypic characterisation of the adaptive and immune response both locally and systemically has been interrogated as a potential source of prognostic biomarkers in this patient population. For example, in patients receiving nivolumab within the CheckMate 040 trial, higher CD3+ or CD8+ T cells infiltrate in pre-treatment tumour samples showed a trend toward improved OS, and increased CD3+ correlated with objective response [15]. Moreover, data from the phase I/II trial testing the association of durvalumab (anti PD-L1) and tremelimumab (anti CTLA-4) at different doses, found that an increase in proliferating peripheral CD8+ T cells at day 15 after treatment start correlated with treatment response [16]. However, while promising, analyses such as these are confined to the experimental setting and are hampered by the lack of biological specimens. The systemic immune milieu is a tumour-agnostic and stage-independent biomarker of poorer outcome [17]. Routinely available measures of systemic inflammation, such as the neutrophil-lymphocyte ratio (NLR) [18], platelet-lymphocyte ratio (PLR) and prognostic nutritional index (PNI) [19], have been previously validated for their survival outcome in HCC prior to clinical use of ICIs [13,20]. These markers have the advantage of being non-invasive, inexpensive, and easy to obtain in routine practice. Before clinical application, there is a need to comprehensively and reproducibly assess the diverse measures of systemic inflammation in patients who receive ICI for HCC. In the present study, we evaluated the prognostic value of NLR, PLR, and PNI in HCC patients treated with ICIs.

## 2. Materials and Methods

Data was collected retrospectively from 472 patients diagnosed with HCC receiving ICI therapy between 2015–2018. Patients were treated in 13 cancer centres across Europe (28%), North America (57%), and Asia (15%). In total, 110 patients had missing blood count and survival data, and were, therefore, excluded from the study, leaving a total of 362 patients for analysis (Figure 1). Clinicopathological characteristics, including patient demographics, aetiology of HCC, complete blood count, maximum diameter of tumour, number of metastases, ECOG performance score, Child-Pugh class, ALBI grade, Barcelona Clinic Liver Cancer (BCLC) stage, and type of immunotherapy, were gathered anonymously from patient records. The NLR was calculated by dividing the total neutrophil count by absolute lymphocyte count, the PLR by dividing the total platelet count by lymphocyte count [21], and the PNI calculated by multiplying albumin (g/l) by absolute lymphocyte count [20]. In line with previous literature, patients with an NLR ≥ 5, PLR ≥ 300 [21], and PNI < 45 [22] were considered to be at high risk of mortality. Inflammatory markers were derived from bloods collected the day of ICIs’ commencement.

Patient demographic and clinicopathological characteristic were reported as median with minimum-maximum ranges for continuous variables and as percentages for qualitative ones. Pearson’s Chi-squared tests were performed to determine associations between variables, with significant associations considered to return a *p*-value <0.05. Kolmogorov-Smirnov tests revealed inflammatory biomarker values to be non-normally distributed. Overall survival (OS), defined as the time between starting ICI therapy until death from any cause or last follow up and progression free survival (PFS), calculated as the time from commencing ICI therapy until the date of progression, last follow up or death were calculated using Kaplan-Meier method, in conjunction with the Log-rank tests to make statistical comparisons. Univariate analysis was carried out to evaluate associations between prognostic factors (inflammatory markers and clinicopathological variables) and survival (OS and PFS) using the Cox regression model, followed by multivariate analysis on variables that were statistically significant. These correlations are reported as hazard ratios (HR) with 95% confidence intervals (CI). Response rate was evaluated according to RECIST v.1.1 criteria. Response parameters include objective response rate (ORR) and disease control rate (DCR). ORR was defined as the number of patients who achieved complete or partial response, while DCR was the number of patients who had responded or maintained a stable disease on ICI therapy. Pearson Chi-squared tests were performed to determine the relationship between inflammatory biomarkers (NLR, PLR and PNI) and these measures of response. The median value of each inflammatory marker was also compared against ORR and DCR using 2 tailed Mann-Whitney tests.

Statistical analysis was performed using SPSS (version 27, IBM Corporation, Armonk, NY, USA) and GraphPad Prism (version 8.2.1 (279), GraphPad Software, Inc San Diego, CA, USA).

## 3. Results

### 3.1. Patient and Disease Characteristics

A total of 362 patients with recorded baseline characteristics were eligible for analysis. The median age of patients at baseline was 65 years (range 15–87) with 284 (78.5%) males and 78 (21.5%) females. Most patients had cirrhosis (*n* = 259, 71.5%), secondary to hepatitis C (*n* = 121, 33.4%), followed by hepatitis B (*n* = 81, 22.4%), alcohol-related cirrhosis (*n* = 81, 22.4%), and non-alcoholic steatohepatitis (NASH) (*n* = 43, 11.9%). Most patients had compensated liver function with 272 (75.1%) falling within Child-Pugh class A criteria and 90 (24.9%) within Child-Pugh class B (B7 *n* = 46, 12.8%; B8 *n* = 32, 8.8%; and B9 *n* = 12, 3.3%). According to ALBI grade, 129 patients (35.6%) were classified as grade 1, 137 as grade 2 (39%), and 96 (25.4%) scored 3 (Table 1). In total, 269 patients (74.3%) had advanced stage HCC (i.e., stage C according to the BCLC staging algorithm), with 80 (22.1%) classified as stage B and 13 (3.6%) stage A, unsuitable for curative surgery or locoregional treatment. Extrahepatic metastases were present in 193 (53.3%) patients and 247 (68.2%) had portal vein thrombosis (PVT); most of the patients had an ECOG performance status (PS) of 0 (*n* = 168, 46.4%) or 1 (*n* = 174, 48.1%), with 17 (4.7%) and 3 (0.8%) falling into ECOG class 2 and 3, respectively. Overall, 80% of the patients were treated with PD-1/PD-L1 monotherapy, with the remaining treated with PD-1/PD-L1 combination therapy with CTLA-4 inhibitors or VEGF pathway inhibitors, treatment was given at standard dose as per indication. ICI treatment was given as second line therapy in 49% of patients (Table 1).

### 3.2. Inflammatory Biomarkers

The median NLR value in the whole cohort was 3.55 (0.06–25.3), PLR 137.32 (0.17–1100), and PNI 40.29 (1.11–1270). An NLR ≥ 5 was recorded in 100 patients (28%) while 53 patients (14.6%) were found to have a PLR ≥ 300 and 207 patients (57%) had a PNI < 45. Significant differences were observed in baseline clinicopathological characteristics between groups stratified by NLR, with patients scoring > 5 reporting higher incidence of PVT (*p* < 0.001), higher ECOG performance status (*p* = 0.009), higher frequency of non-viral etiology (*p* = 0.04), and more advanced BCLC stage (*p* = 0.029). Patients stratified by PLR varied significantly in ECOG performance status (*p* = 0.011), BCLC stage (*p* < 0.001), Child-Pugh class (*p* = 0.025), and for the presence of PVT (*p* = 0.004). Patient groups split by PNI status exhibited differences in Child-Pugh class (*p* < 0.001), ECOG performance status (*p* < 0.001), BCLC stage (*p* = 0.002), and the number of patients with PVT (*p* = 0.001) (Table 2). At data cut-off, 54% of the patients had deceased, with a median OS of 9 months (range 0.43–53.3 months). Significantly shorter median OS was observed in patients with NLR ≥ 5 (7.7 vs. 17.6 months, HR 2.29, 95%CI 1.70–3.09, *p* < 0.0001) (Figure 2A), PLR ≥ 300 (6.7 vs. 16.5 months, HR 2.45, 95%CI 1.70–3.52) (Figure 2C), and PNI <45 (10.8 vs. 17.7 months, HR 1.60, 95%CI 1.23–2.18, *p* = 0.018) (Figure 2E). In univariate analysis, also the presence of PVT (HR 1.78, 95%CI 1.34–2.38, *p* < 0.001) and Child-Pugh class (B vs. A) correlated with poorer OS (HR 1.81, 95%CI 1.33–2.46, *p* < 0.001) (Table 3). NLR, PLR, presence of PVT, PNI, and Child-Pugh class were selected in the multivariate analysis (Table 3). NLR ≥ 5 and PLR ≥ 300 remained independent prognostic factors for OS (HR 1.73, 95%CI 1.23–2.42, *p* = 0.002; HR 1.60, 95%CI 1.6–2.40, *p* = 0.020), as did the presence of PVT (HR 1.49, 95%CI 1.02–2.02, *p* = 0.010) and Child-Pugh class B vs. A (HR 1.62, 95%CI 1.17–2.25, *p* = 0.004). No significant association was observed between OS and PNI (HR 0.99, 95%CI 0.71–1.37, *p* = 0.940).

The median PFS was 3.5 months (0.5–42.2). Patients with NLR ≥ 5 had a significantly shorter PFS compared to the counterpart with lower NLR (2.1 vs. 3.8 months, HR 1.83, 95%CI 1.32–2.55 *p* = 0.03) (Figure 2B); similarly, patients with PLR ≥ 300 reported shorter PFS (1.8 vs. 3.7 months, HR 2.07, 95%CI 1.27–3–38 *p* = 0.0006) (Figure 2D). PFS did not significantly differ according to PNI (2.5 vs. 4 months, HR 1.28, 95% CI 0.91–1.65 *p* = 0.17) (Figure 2F). The other parameter associated with worse PFS in univariate analysis was the presence of PVT (2.7 vs. 3.7 months, HR 1.53, 95%CI 1.04–2.24, *p* = 0.030) (Table 4). Only PLR ≥ 300 remained a predictor of worse PFS in multivariate analysis (HR 1.99, 95%C 1.11–3.49, *p* = 0.021) (Table 4). Out of 362 patients, 343 were evaluable for response. Overall, ORR was 19% and DCR was 59%, at the time of data analysis, 78% of patients had experienced progression. No significant associations were observed between response and patient baseline characteristics including age, presence of PVT, Child-Pugh class, BCLC stage, and ECOG performance status. Patients with NLR ≥ 5 had a significantly worse ORR compared to those with NLR < 5 (12% vs. 22%, *p* = 0.034) (Figure 3A). PNI < 45 correlated with worse DCR compared to PNI ≥ 45 (52% vs. 66%, *p* = 0.014) (Figure 3D) and showed a trend toward worse ORR (24% vs. 16%, *p* = 0.063) (Figure 3C); no statistical difference in terms of DCR was observed in NLR high vs. NLR low patients (60% vs. 53%, *p* = 0.230) (Figure 3B) and in ORR or DCR according to PLR status (Figure 3D,F).

Patients who were refractory to ICI therapy (i.e., those who achieved PD as best overall response) had a significantly higher NLR compared to those who responded or maintained a stable disease (3.83 vs. 3.37, *p* = 0.046). PNI was significantly lower in patients who had stable or progressive disease, compared to those who demonstrated a partial or complete response to ICIs (38.72 vs. 46.65, *p* = 0.021). PLR did not vary significantly according to response status.

## 4. Discussion

The introduction of immune checkpoint inhibition has substantially changed the medical management of HCC [23]. However, with ICI monotherapy, response rates remain as low as 20% [4], and clinical predictive biomarkers are urgently required to adequately stratify patients based on their potential to respond to ICIs. In this study, we have demonstrated that serum inflammatory biomarkers, which have the advantage of being un-expensive, easily reproducible, and broadly available, are useful to identify, among HCC patients receiving ICIs, those with worse prognosis. In particular, our data show that systemic inflammation measured by NLR is associated with worse response and survival and it is also a prognostic factor for OS independent of common clinicopathological characteristics, as demonstrated by results of the multivariate analysis. Moreover, PLR, despite not being associated with response, was demonstrated to be prognostic for both OS and PFS. This is in keeping with previous studies reporting similar results in patients receiving nivolumab monotherapy [15,21]; however, this is the only study to investigate NLR and PLR in a cohort of patients treated with different types of ICIs. Furthermore, we also tested the association of PNI with outcomes and this is the first paper to report about the prognostic role of PNI in HCC patients receiving ICI.

In our population, PNI did not retain a prognostic significance once adjusted for possible cofounding variables and this is probably due to the intimate correlation existing between PNI and residual liver function. In fact, albumin level is strictly dependent on liver synthetic ability, and, therefore, PNI should be interpreted with caution in the case of deranged liver function. Our results about the prognostic role of PLR are in keeping with data reported by Dharmapuri [21], showing a significant correlation between PLR and overall survival. Even if, in HCC patients, thrombocytopenia is often an indicator of hypersplenism secondary to portal hypertension, which can by itself hamper the prognosis, increased PLT reflected by higher PLR retained a prognostic significance. With new ICIs combination therapies being tested in phase III trials, the first line scenario of advanced HCC is expected to rapidly expand and patient selection will become pivotal to choose the best treatment for every patient. We believe that in this context, where other biomarkers have failed, inflammation markers deserve further investigations. In fact, predictive biomarkers typically adopted to stratify ICIs patients in other cancers have produced poor results in HCC. In particular, the level of PD-L1 expression on tumour infiltrating cells and/or tumour cells has a prognostic role in HCC [24], but evidence about its predictive value in patients receiving ICIs is lacking. In fact, PD-L1 is not adopted in clinical practice, or to stratify patients in ICI trials. The two main limitations in its use depend on the spatial and temporal heterogeneity of its expression, and on the absence of a standard method for its detection and interpretation [11]. Similarly, high tumour mutational burden and micro-satellite instability are rarely detected in HCC and are, therefore, not clinically useful [25]. Genomic studies have revealed the presence of genetic traits [26] which could be implicated in ICI resistance, and a classification into immune classes according to molecular features has been proposed [27]; however, prospective data to translate this data into routine practice are currently lacking. It should also be highlighted that biopsy is not mandated to diagnose HCC when radiologic criteria are satisfied and, on one hand, this aspect limits the availability of biological specimens for experimental purposes and on the other hand, suggests the need to search for non-invasive biomarkers.

NLR, PLR, and PNI have the advantage of being easily derived from routine blood tests and their interpretation does not require specific training. The rationale to investigate these markers stems from their ability to mirror the systemic inflammatory status which often results in increased relative levels of neutrophils and platelets, as reflected by high NLR and PLR. Neutrophils and platelets have been implicated in cancer immune-escape and progression trough cytokines productions (e.g., IL-18, VEGF, and PDGF) [17]. Furthermore, circulating neutrophils have been reported to promote cancer growth and progression trough neutrophiles extra-cellular traps formation (NET) [28], and tumour-associated neutrophiles have been described to foster cancer progression via HCC stem-like cells stimulation [29]. Immature neutrophils are a subset of cells often classified as myeloid derived suppressor cells (MDSCs), found in high numbers in HCC and other cancers, and preclinical evidence suggests that targeting MDSCs has the potential to revert the immunosuppressive microenvironment typical of inflammation-based cancers [30,31].

To our knowledge, this is the first study to evaluate the prognostic value of NLR, PLR, and PNI in a large cohort of patients receiving ICIs; however, our findings need to be validated in a prospective manner and the correlation between these markers and other markers of inflammation (e.g., peripheral T cells phenotype) requires further investigation. Along with the retrospective nature, one of the major limitations to our study is the absence of biomarkers analysis at different time-points during ICIs therapy and the lack of data regarding patients’ pre-existing co-morbidities, which could contribute to systemic inflammation and act as confounding factors (e.g., metabolic syndrome). Moreover, in the absence of a control cohort, our study does not provide predictive information; however, in the presence of a significant correlation between NLR and response, a direct interaction with the treatment could be hypothesised.

## 5. Conclusions

In conclusion, our findings, coming from a large multi-centric cohort, highlight the prognostic role of NLR and PLR in ICIs recipients. With the emergence of new therapeutic strategies, the biological mechanism underlying the association between systemic inflammation and anti-cancer response requires specific evaluation. Further studies are warranted to define the exact predictive meaning of these markers and to unveil the complex interplay between systemic inflammation and immune system in the context of ICIs therapy.

## Figures and Tables

**Figure 1 cancers-14-00186-f001:**
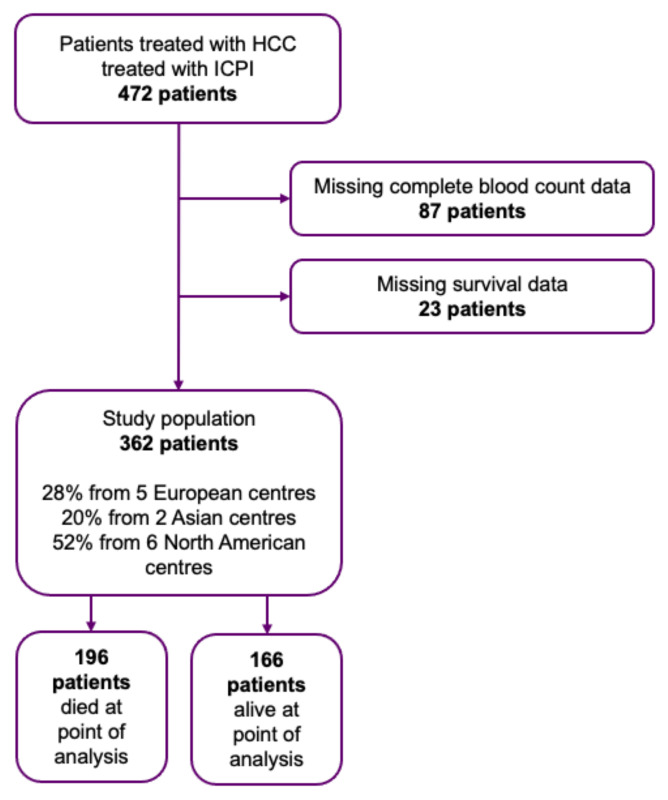
Study population after excluding patients not eligible for the final analysis.

**Figure 2 cancers-14-00186-f002:**
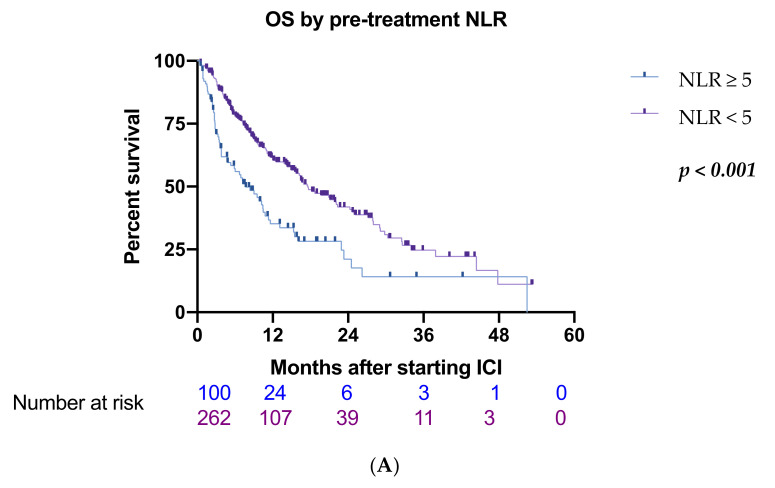
Kaplan-Meier curves for progression free survival (PFS) and overall survival (OS) according to inflammatory markers. (**A**) OS according to neutrophil to lymphocytes ratio (NLR), (**B**) PFS according to NLR, (**C**) OS according to platelet to lymphocytes ratio (PLR), (**D**) PFS according to PLR, (**E**) OS according to prognostic nutritional index (PNI), (**F**) PFS according to PNI.

**Figure 3 cancers-14-00186-f003:**
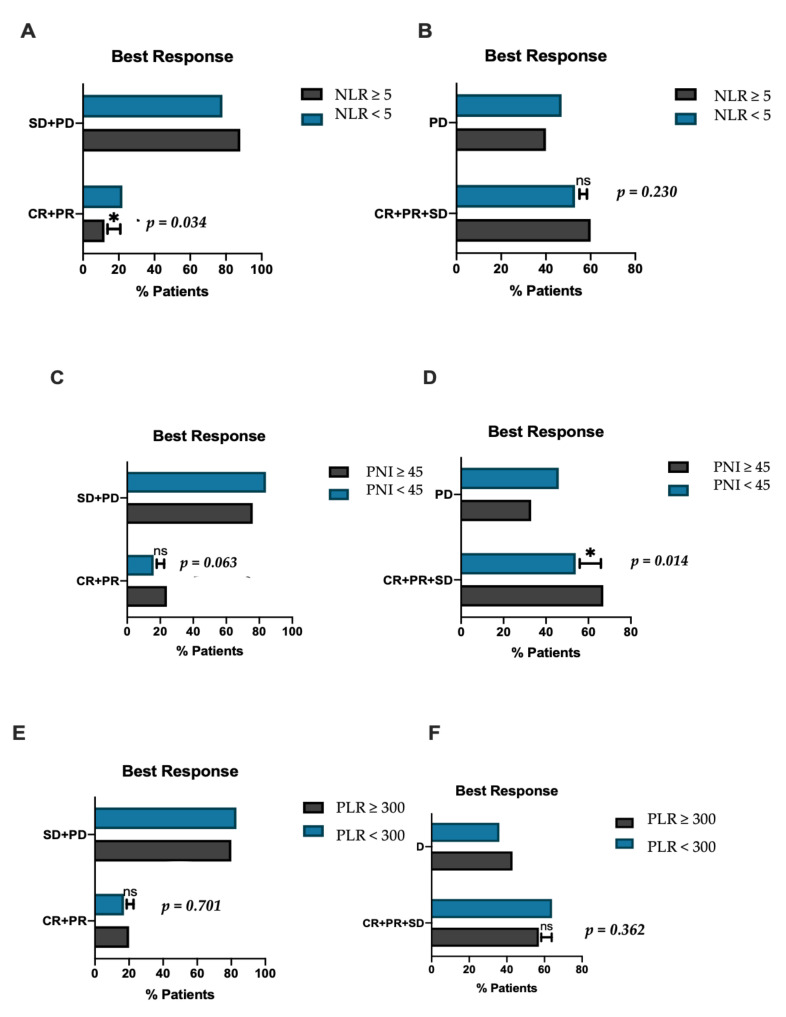
Histograms demonstrating the differences in inflammatory biomarkers according to response. (**A**) Proportion of patients reporting objective response (CR + PR) according to neutrophil to lymphocytes ratio (NLR), (**B**) proportion of patients reporting disease control (CR + PR + SD) according to NLR, (**C**) proportion of patients reporting objective response (CR + PR) according to prognostic nutritional index (PNI), (**D**) proportion of patients reporting disease control (CR + PR + SD) according to PNI, (**E**) proportion of patients reporting objective response (CR + PR) according platelet to lymphocytes ratio (PLR), (**F**) proportion of patients reporting disease control (CR + PR + SD) according to PLR. NS: non-significant; * significant; SD: stable disease; PD: progressive disease; CR: complete response; PR: partial response.

**Table 1 cancers-14-00186-t001:** Clinicopathological features at baseline.

Gender	
Male	284 (78.5)
Female	78 (21.5)
Age	Median (LQ-UQ) = 65 (58–70)
<65	180 (49.7)
≥65	182 (50.3)
Aetiology	
HBV	81 (22.4)
HCV	121 (33.4)
Alcohol induced	81 (22.4)
NASH	43 (11.9)
Other	36 (9.94)
Cirrhosis	
Present	259 (71.5)
Absent	103 (28.5)
Portal vein thrombosis	
Present	247 (68.2)
Absent	115 (31.5)
Child-Pugh Class	
A	272 (75.1)
B	90 (24.9)
ALBI grade	
1	129 (35.6)
2	137 (29)
3	96 (25.4)
ECOG Performance status	
0	168 (46.4)
1	174 (48.1)
2	17 (4.7)
3	3 (0.8)
Barcelona Clinic Liver Cancer stage	
A	13 (3.6)
B	80 (22.1)
C	269 (74.3)
Mean tumour diameter	Median (LQ-UQ) = 6.0 (3.0 −10.8)
Extrahepatic metastasis	
Present	193 (53.3)
Absent	169 (46.7)
Immunotherapy	
Nivolumab	218 (60.2)
Pembrolizumab	45 (12.4)
Ipilimumab	1 (0.3)
Ipilimumab/Nivolumab	13 (3.6)
Avelumab	1 (0.3)
Atezolizumab	11 (3.0)
Durvalumab	8 (2.2)
Other PD-1 single agents	13 (3.6)
PD-1, CTLA-4 combination	14 (3.9)
PD-1, TKI combination	24 (6.6)
Other PD-1 combinations	14 (3.9)

**Table 2 cancers-14-00186-t002:** Relationship between inflammatory status and baseline clinicopathological characteristics. NLR: neutrophil to lymphocytes ratio; PLR: platelet to lymphocytes ratio; PNI: prognostic nutritional index; BCLC: Barcelona Clinic Liver Cancer; ALBI: albumin to bilirubin; * *p* < 0.05; ** *p* < 0.01; *** *p* < 0.001.

Variable	NLR		*p*-Value	PLR		*p*-Value	PNI		*p*-Value
	<5	≥5		<300	≥300		<45	≥45	
**Portal-vein thrombosis** Absent/ Present	194/68 (74%)(26%)	53/47 (53%)(47%)	≤0.001 ***	217/92 (70.2%) (29.8%)	26/27 (49%)(51%)	0.004 **	126/81 (61%)(39%)	121/34 (78%) (22%)	0.001 **
**Child-Pugh class** A/B	202/60 (77%)(23%)	70/30 (70%)(30%)	0.162	239/70 (77.3%) (22.7%)	33/20 (62.3%) (37.7%)	0.025 *	135/72 (65%)(35%)	137/18 (88%) (12%)	<0.001 ***
**ECOG performance status** 0/1/2/3	135/116/9/2 (51.5%) (44.2%) (3.4%)(0.9%)	33/58/8/1 (33%)(58%) (8%)(1%)	0.009 **	152/143/14/0 (49%) (46%) (5%)	16/29/8/0 (30.2%) (54.7%) (5.1%)	0.011 *	72/117/16/2 (34.8)(56%) (7.7%) (9.2%)	96/57/1/1 (61.9%) (36.7%) (0.7%)	<0.001 ***
**BCLC stage** A/B/C	10/67/185 (3.8%) (25.6%) (70.6%)	3/13/84 (3%)(13%) (84%)	0.029 *	7/75/227 (2.3%) (24.2%) (73.5%)	3/8/42 (5.6%) (15%) (79.4%)	<0.001 ***	11/34/162 (5.3%) (16.4%) (78.3%)	2/46/107 (1.4%) (29.6%) (69%)	0.002 **
**ALBI grade** 1/2/3	99/90/73 (37.8%)(34%) (28.2%)	31/49/20 (31%)(49%) (20%)	0.334	114/74/121 (36.8%) (24%) (39.2%)	18/28/7 (33.9%) (52.8%) (13.3%)	0.758	61/92/54 (29.5%) (44.4%) (26.1%)	71/38/46 (45.8%) (24.5%) (29.7%)	0.279
**Extrahepatic spread** Absent/ Present	127/135 (48%)(52%)	42/58 (42%)(58%)	0.270	146/163 (47.2%) (52.8%)	23/30 (43.4%) (56.6%)	0.656	94/113 (45%)(55%)	75/80 (48%) (52%)	0.574
**Aetiology** **Viral/** **Non-viral**	180/107 (62%)(38%)	44/45 (49%)(51%)	0.04 *	169/144 (54%)(46%)	23/24 (49%)(51%)	0.53	115/91 (56%)(44%)	85/69 (55%)(45%)	0.92

**Table 3 cancers-14-00186-t003:** Univariate and multivariate Cox regression model assessing for overall survival. NLR: neutrophil to lymphocytes ratio; PLR: platelet to lymphocytes ratio; PNI: prognostic nutritional index; PVT: portal vein thrombosis; HCC: hepatocellular carcinoma; BCLC: Barcelona Clinic Liver Cancer; ALBI: albumin to bilirubin; * *p* < 0.05; ** *p* < 0.01; **** p* < 0.001.

Prognostic Factor		Univariate Analysis		Multivariate Analysis	
	*n* = 362	HR (95% CI)	*p* Value	HR (95%)	*p* Value
**NLR**					
≥5/<5	100/262	1.95 (1.45–2.64)	<0.001 ***	1.73 (1.23–2.42)	0.002 **
**PLR**					
≥300/<300	53/309	2.05 (1.42–2.98)	<0.001 ***	1.60 (1.6–2.40)	0.020 *
**PNI**					
≥45/<45	207/155	0.71 (0.53–0.94)	0.018 *	0.99 (0.71–1.37)	0.940
**PVT**					
Present/Absent	247/115	1.78 (1.34–2.38)	<0.001 ***	1.49 (1.02–2.02)	0.010 *
**ECOG performance score**					
0–1/2–3	342/20	1.49 (0.83–2.67)	0.186		
**ALBI grade**					
1/2–3	129/233	1.30 (0.90–1.89)	0.151		
**BCLC stage**					
C/A-B	269/93	1.19 (0.85–1.64)	0.309		
**Child-Pugh class**					
B/A	90/272	1.81 (1.33–2.46)	<0.001 ***	1.62 (1.17–2.25)	0.004 **
**Extrahepatic metastasis**					
Present/Absent	193/169	1.17 (0.88–1.55)	0.275		
			HCC Aetiology		
**Viral/Non-viral**	197/164	0.93 (0.70−1.24)	0.620		

**Table 4 cancers-14-00186-t004:** Univariate and multivariate Cox regression model for progression free survival. NLR: neutrophil to lymphocytes ratio; PLR: platelet to lymphocytes ratio; PNI: prognostic nutritional index; PVT: portal vein thrombosis; HCC: hepatocellular carcinoma; BCLC: Barcelona Clinic Liver Cancer; ALBI: albumin to bilirubin; * *p* < 0.05; ** *p* < 0.01.

Prognostic Factor		Univariate Analysis		Multivariate Analysis	
	*n* = 362	HR (95% CI)	*p* Value	HR (95%)	*p* Value
**NLR**					
≥5/<5	100/262	1.54 (1.03–2.30)	0.036 *	1.21 (0.82–1.78)	0.331
**PLR**					
≥ 300/<300	53/309	2.33 (1.41–3.83)	0.001 **	1.99 (1.11–3.49)	0.021 *
**PNI**					
≥ 45/<45	207/155	0.86 (0.60–1.24)	0.423		
**PVT**					
Present/Absent	247/115	1.53 (1.04–2.24)	0.030 *	1.12 (0.81–1.58)	0.480
**ECOG performance score**					
0–1/2–3	342/20	1.18 (0.58–2.43)	0.649		
**ALBI grade**					
1/2–3	129/233	0.72 (0.50–1.04)	0.091		
**Child-Pugh class**					
B/A	90/272	1.39 (0.92–2.09)	0.115		
**Extrahepatic metastasis**					
Present/Absent	193/169	0.97 (0.67–1.39)	0.855		
			HCC Aetiology		
**Viral/Non-viral**	197/164	0.85 (0–62-1.15)	0.290		

## Data Availability

The data presented in this study are available on request from the corresponding author.

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
