# Peer review of "The Systemic Inflammatory Response Identifies Patients with Adverse Clinical Outcome from Immunotherapy in Hepatocellular Carcinoma"

_cancers, 2021, doi:10.3390/cancers14010186_

Round 1

Reviewer 1 Report

Manuscript Number: cancers-1441759

Comments to author::

Major:

  1. Have baseline data of type 2 DM, lipid profile, BMI and hypertension in those hepatocellular carcinoma patients.
  2. Could you try to evaluate relationship between baseline data of type 2 DM, lipid profile, BMI, and hypertension and inflammatory status?
  3. Had all HCC patients be treated with standard dose of immune checkpoint drugs.

Reviewer 2 Report

The work by Muhammed et al. was focused on the evaluation of the systemic inflammatory response as a clinical outcome in patients with hepatocellular carcinoma undergoing immunotherapy. By conducting a retrospective study including 362 patients with HCC receiving immune checkpoint inhibitors (ICI), the authors observed that patients with NLR>5 displayed a shorter OS and a lower ORR and PFS. Furthermore, they observed that NLR is an independent prognostic factor for OS while PLR independently predicted OS and PFS. Overall, this work provides novel relevant findings in the field. The work was nicely conducted and the analysis are robust. Only small concerns should be addressed in order to increase the overall quality of this work.

1 – Are there any differences of the prognostic value of these parameters according HCC etiology?

2 – In the multivariate analysis, it is not clear if the authors also included BLCL stage. Are these parameters independent on tumor stage?

3 – What about the other prognostic factors mentioned in introduction? Do the authors confirm their prognostic value? Are the new ones better predictors of prognosis compared to, for instance, CD3 and CD8 cells?
